# Semi-solid prodrug nanoparticles for long-acting delivery of water-soluble antiretroviral drugs within combination HIV therapies

James J. Hobson[1], Amer Al-khouja[2], Paul Curley[3], David Meyers[2], Charles Flexner[2,4], Marco Siccardi[3], Andrew Owen 🔘 [3], Caren Freel Meyers[2] & Steve P. Rannard 🔘 [1]

The increasing global prevalence of human immunodeficiency virus (HIV) is estimated at 36.7 million people currently infected. Lifelong antiretroviral (ARV) drug combination dosing allows management as a chronic condition by suppressing circulating viral load to allow for a near-normal life; however, the daily burden of oral administration may lead to non-adherence and drug resistance development. Long-acting (LA) depot injections of nanomilled poorly water-soluble ARVs have shown highly promising clinical results with drug exposure largely maintained over months after a single injection. ARV oral combinations rely on water-soluble backbone drugs which are not compatible with nanomilling. Here, we evaluate a unique prodrug/nanoparticle formation strategy to facilitate semi-solid prodrug nanoparticles (SSPNs) of the highly water-soluble nucleoside reverse transcriptase inhibitor (NRTI) emtricitabine (FTC), and injectable aqueous nanodispersions; in vitro to in vivo extrapolation (IVIVE) modelling predicts sustained prodrug release, with activation in relevant biological environments, representing a first step towards complete injectable LA regimens containing NRTIs.

[1] Department of Chemistry, University of Liverpool, Crown Street, Liverpool L69 7ZD, UK. [2] Department of Pharmacology and Molecular Sciences, The Johns Hopkins University School of Medicine, 725 North Wolfe St., Baltimore, MD 21205, USA. [3] Department of Molecular and Clinical Pharmacology, University of Liverpool, Block H, 70 Pembroke Place, Liverpool L69 3GF, UK. [4] Department of Medicine, The Johns Hopkins University School of Medicine, 575 Osler Building, 600N. Wolfe St., Baltimore, MD 21287, USA. These authors contributed equally: James J. Hobson, Amer Al-khouja. These authors jointly supervised this work: Andrew Owen, Caren Freel Meyers, Steve P. Rannard. Correspondence and requests for materials should be addressed to A.O. (email: aowen@liverpool.ac.uk) or to C.F.M. (email: cmeyers@jhmi.edu) or to S.P.R. (email: srannard@liverpool.ac.uk)

Chronic oral dosing of solid therapeutic or prophylactic medicines is known to lead to pill fatigue[1], even in healthy patient populations taking nutraceuticals. Where adherence to therapy is a key determinant of clinical outcomes, the impact of non-adherence can be significant to the individual and the wider community. Individual non-adherence can lead to drug resistance and the transmission of resistant virus throughout patient populations that are largely adherent to therapy[2]. Therapeutic failure has been estimated in 8% of treatment-naive and 33% of treatment-experienced human immunodeficiency virus (HIV) patients. Post- and pre-exposure prophylaxis (PrEP) is also a globally active research theme aiming at controlling transmission rates; however, suboptimal adherence also places patients at risk of low protection from PrEP[3]. Considerable efforts have reduced dosing frequencies to once-daily oral regimens in many cases, while co-formulation has reduced the daily number of pills by offering fixed-dosed combination tablets. Nevertheless, pill fatigue remains a problem, and major efforts are underway to develop long-acting (LA) antiretrovirals (ARVs) that require less frequent dosing to minimise non-adherence.

There are six classes of ARVs, with nucleoside reverse transcriptase inhibitors (NRTIs) as the backbone therapy critical to clinically successful combinations for treatment and the main pharmacological component of PrEP strategies[4]. Successful human trials have been widely reported for the subcutaneous and intramuscular (IM) LA depot injections of two ARVs, the non-nucleoside reverse transcriptase inhibitor rilpivirine and an integrase inhibitor, cabotegravir[5–7]. Individually they represent monotherapy, and oral NRTI backbone therapy is required when utilising these injections independently. Studies of rilpivirine/cabotegravir combination injections suggest such LA regimens may be as effective in suppression of HIV-1 viral replication as an oral three-drug combination (cabotegravir/abacavir/lamivudine) with very high tolerance and patient acceptance[8]. This combination, representing a dual-nanoparticle LA option, is exciting; however, two-drug therapy is not yet a standard of care for HIV treatment[9], and as the only potential LA strategy for treatment and PrEP, cabotegravir/rilpivirine will not address resistance or intolerance[10]. Further, vulnerable populations (neonates, children, adolescents, and pregnant women) cannot use this combination as safety data and clear dosing strategies are still lacking. Multiple LA therapies would be available to address these issues if rilpivirine and cabotegravir could be combined with NRTIs in nanodispersed form.

The most successful commercial route for solid drug nanoparticle (SDN) manufacture is nanomilling, used for LA cabotegravir and rilpivirine, but other attrition-based techniques, such as high-pressure homogenisation, have been used to reduce large particles into the sub-micron range. These approaches for SDN manufacture require the active pharmaceutical ingredient (API) to have low water solubility and are thus not applicable for developing LA formulations of water-soluble NRTIs. In recent years, emulsion-templated freeze drying (ETFD)[11] has been shown to be a valuable addition to SDN manufacturing options, allowing non-attrition formation routes to be studied for APIs with diverse chemical and physical properties (including varying water solubility, low melting point, and semi-solid compounds), and large libraries of potential SDN-derived nanomedicines can be rapidly developed and screened to identify candidates with clinical potential. ETFD has also been progressed to emulsion spray drying to allow scaling of options under clinically controlled good manufacturing practice (GMP) conditions leading to human trials[12]. Emtricitabine (FTC) is widely used in a range of HIV regimens such as Truvada® (FTC/tenofovir disoproxil fumarate (TDF)), Atripla® (FTC/TDF/efavirenz), Symtuza™ (darunavir/cobicistat/FTC/tenofovir alafenamide) and Stribild® (FTC/TDF/cobicistat/ elvitegravir) and has a reported water solubility of 112 mg mL$^{-1}$ at 25 °C[13]. We hypothesised that reversible masking of FTC to incorporate a range of hydrophobic alkyl carbonate and/or carbamate groups would generate prodrugs with poor water solubility that could be readily processed using ETFD approaches. Due to the potential for a reduction of melting point when using such a strategy (as seen with paliperidone palmitate), the prodrugs may become incompatible with conventional nanoparticle formation techniques such as nanomilling. Given the potential stability of carbamate functional groups in target cells, intact prodrugs are not expected to exhibit antiviral activity. However, the formation of stabilised prodrug nanoparticles, capable of releasing prodrug molecules that are activated under relevant biological conditions, could lead to candidates for HIV combination depot applications (Fig. 1).

Here we demonstrate the unique potential of ETFD to develop libraries of candidate therapies from low melting point FTC prodrugs shown to undergo bioactivation in physiological conditions relevant to long-acting depot administration and apply appropriate models to elucidate the potential benefits that may be observed in future human studies.

## Results

**Prodrug synthesis and activation studies.** Inspired by the clinically utilised capecitabine, a 5-fluorouracil pentyl carbamate prodrug activated by human carboxylesterase in the liver[14], eight FTC prodrugs modified at the 5'-hydroxyl and amino groups (Fig. 1, 1–8) were generated in single step syntheses from FTC through the facile reaction of both the amine and hydroxyl functional groups with alkyl chloroformates of varying chain length (Supplementary Methods and Supplementary Figures 1–16). Selective hydrolysis of the 5'-carbonate groups of 1–8 afforded FTC carbamate prodrugs 9–16 to give a total of 16 prodrug analogues spanning a range of logP and physical properties[14–16]. The prodrugs within the library were all low-melting, semi-solid compounds, otherwise incompatible with conventional nanomilling technologies, exhibiting increasing hydrophobicity with increasing alkyl chain length (calculated logP (clogP) range of −0.16 to 5.50). The similarity of the approach to the masking of capecitabine also considerably de-risks this strategy with respect to potential safety concerns and future translation to human studies.

Individually, the 16 prodrugs were assessed using high-performance liquid chromatography (HPLC)-based methods for activation to FTC within human plasma, and human liver and muscle subcellular S9 fractions (Fig. 2; Supplementary Figures 17–22 and Supplementary Table 1). As expected, rapid enzyme-catalysed hydrolysis of the 5'-carbonate group of 1–8 was observed, evident from the short plasma half-lives (<12 h) and relatively rapid initial hydrolysis rates in liver S9 and muscle S9 fractions (evaluated over a timeframe in which the carbamate masking group remains intact). Enzyme-catalysed hydrolysis of the carbamate moieties of 9–16 was studied under the same physiological conditions and found to be significantly more stable compared to prodrugs 1–8 (Supplementary Figures 19 & 21). Although chemically similar to capecitabine at the carbamate moiety, the plasma half-lives of longer-chain carbamates 12–16 were generally shorter. Initial hydrolysis rates in liver and muscle S9 fractions were within a sixfold range of capecitabine as measured in our hands, with the longer $C_5$ to $C_8$ analogues (13–16) undergoing hydrolysis most efficiently in liver S9 (Supplementary Figure 21). Interestingly, the plasma half-lives of 5'-carbonate analogues (hydrolysis of 5'-carbonate moiety) were shortest for the $C_2$ and $C_3$ analogues (2 and 3); prodrug stability in plasma increased with longer carbonate alkyl chains

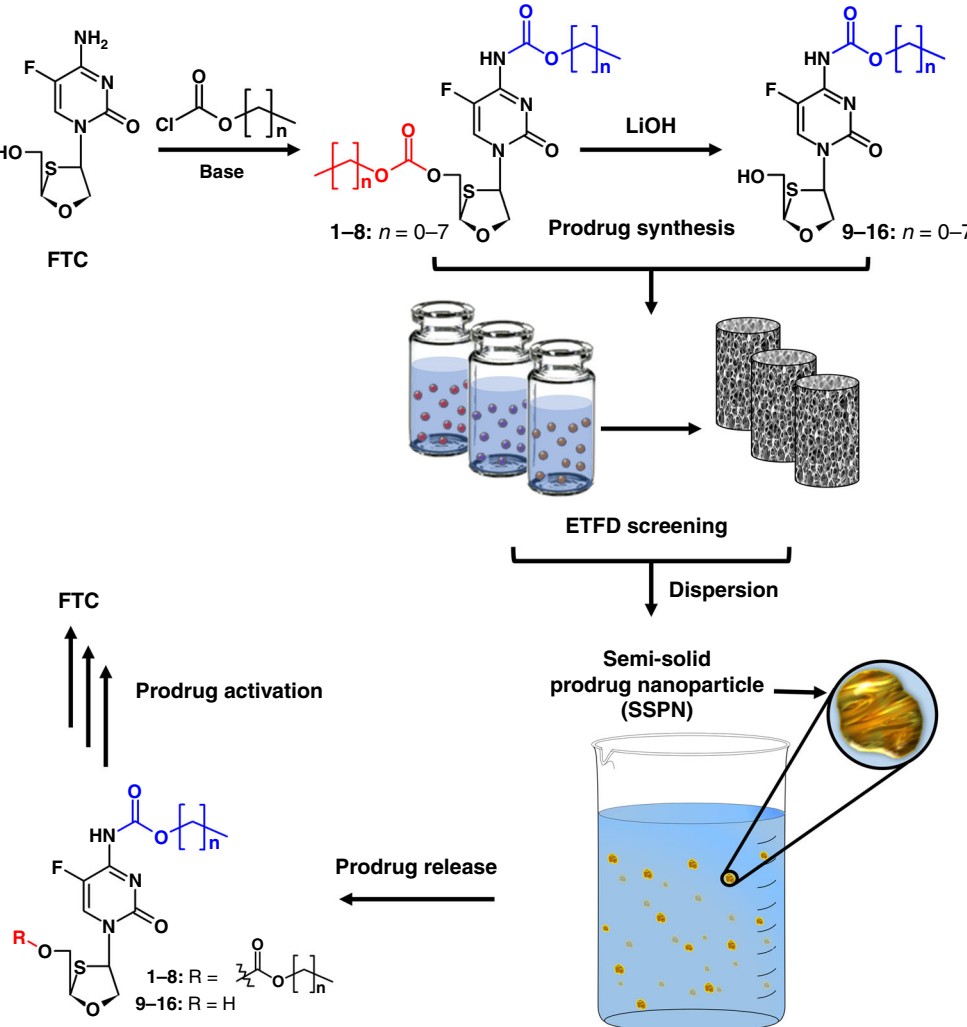

**Fig. 1** Schematic overview of long-acting semi-solid prodrug nanoparticle (SSPN) strategy. Emulsion-templated freeze drying (ETFD) screening is applied to a range of emtrictabine (FTC) prodrugs bearing bio-reversible carbonate and carbamate masking groups. SSPNs are released into aqueous media after rapid dispersion of ETFD monoliths into water, resulting in a dispersion of SSPNs slowly releasing prodrug under physiologically relevant conditions and subsequent bioactivation to release the parent FTC

($C_4$ to $C_8$, Fig. 2b). 5'-Carbonate hydrolysis rates in muscle and liver S9 reached a maximum with the $C_3$ analogue (**3**, Fig. 2c, d); a marked decrease in hydrolysis rate was observed with shorter or longer carbon chain lengths. In all cases, the hydrolysis of the carbonate moiety of **1–8** was considerably faster in liver than muscle.

**Screening and formation of semi-solid prodrug nanoparticles**. The ability to generate nanoparticles from the low-melting point (below ambient temperature) prodrugs was assessed using ETFD, with the products being more correctly termed semi-solid pro-drug nanoparticles (SSPNs) rather than conventional SDNs. Seven polymer and six surfactant excipients (Supplementary Table 2) were chosen from the U.S. Food and Drug Administration Center for Drug Evaluation and Research Inactive Ingredient Database[17]; initial SSPN formation studies were conducted using chloroform solutions of each prodrug, acting as the dispersed phase of the emulsion, and binary polymer/surfactant mixtures within the aqueous continuous phase (1:4 oil/water ratio). This generated 16 libraries, each containing 42 candidate SSPNs (672 total SSPNs) with a 10 wt% drug loading relative to water-soluble excipients in the final lyophilised monolith

(Supplementary Figures 24–41). In this accelerated screening approach, each solid monolith was re-dispersed in water and evaluated using dynamic light scattering (DLS) to establish quality of re-dispersion (target 1 mg mL$^{-1}$), $z$-average diameter (target <1000 nm to match existing LA paradigms), reproducibility over three scans (<5%) to indicate stability, and polydispersity index (target <0.4) to indicate uniformity of SSPNs (Fig. 3).

The 672 candidates were ranked against these targets and a clear trend of the number of hits was observed across the 16 prodrugs; relatively few hits were observed for compounds **1–3** and **9–16**, with over half of the potential candidates producing hits for compounds **4–8** bearing masking groups at both 5'-hydroxyl and amino groups (Fig. 3a). Of interest was the lack of SSPNs formed from the mono-substituted carbamate-modified FTC prodrug series (the capecitabine analogue structures) and the dramatic increase in hits observed for fully masked prodrugs **1–8**; this trend did not entirely correspond with increasing clog$P$ values as compounds **1–5** spanned similar clog$P$ values as **11–16** but with considerably different outcomes from the ETFD process. All of the hits displayed narrow, monomodal particle size distributions (Fig. 3b; Supplementary Figures 42–44), with $z$-average diameters ranging from 15 to 496 nm and

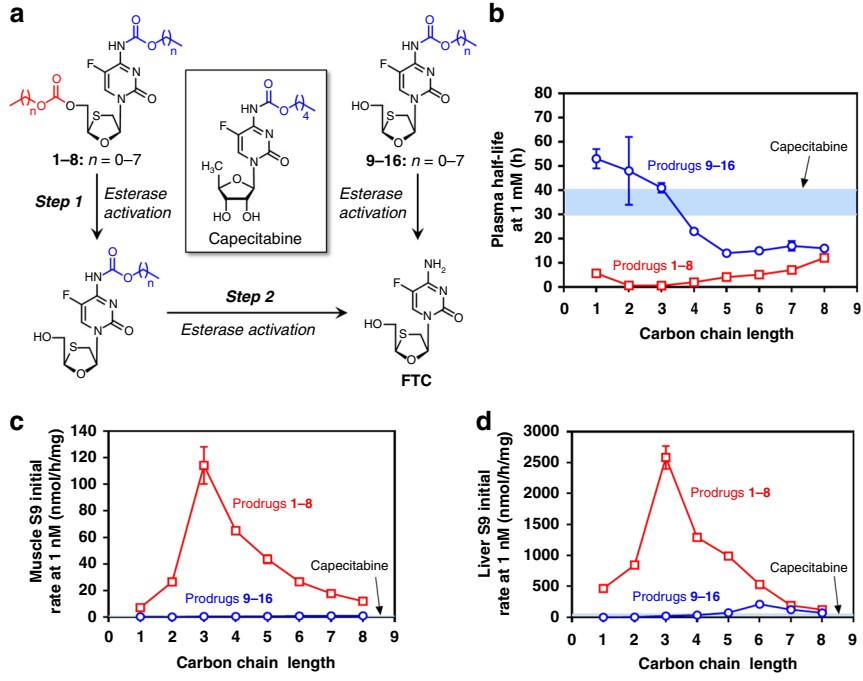

**Fig. 2** In vitro metabolism of emtricitabin (FTC) prodrugs under physiologically relevant conditions. **a** Initial rates of esterase-mediated hydrolysis of 5'-carbonate (1 mM, **1–8**) to generate corresponding carbamates, and initial rates of hydrolysis of carbamates (1 mM, **9–16**), resembling capecitabine, to generate FTC was monitored in undiluted human plasma (**b**), muscle S9 (**c**), and liver S9 (**d**) fractions. The light blue band represents the mean ± SD for carbamate cleavage from capecitabine

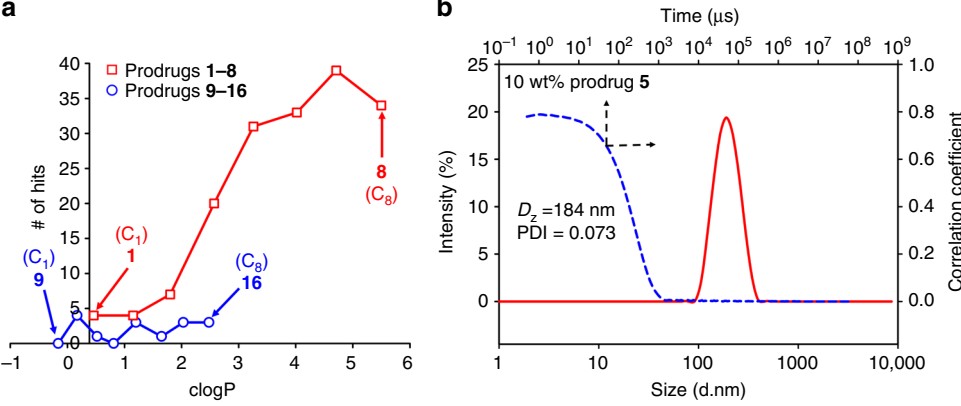

**Fig. 3** Success of emulsion-templated freeze drying screening to form semi-solid prodrug nanoparticles (SSPNs). **a** Relationship between calculated log$P$ (clog$P$) and number of hits identified at 10 wt% prodrug loading relative to water-soluble excipients for each prodrug candidate, and **b** dynamic light scattering of a representative SSPN dispersion in water (1 mg mL$^{-1}$; 10 wt% prodrug **5**, 60 wt% PVP K30, 30 wt% Tween 20)

polydispersity values varying from 0.010 to 0.297 across the two classes of prodrugs. The ten hits considered worthy of further investigation were judged as having monomodal, near-monodisperse (polydispersity <0.2) distributions with z-average diameters <250 nm; these criteria were selected as uniformity was expected to confer reproducible drug release.

A low loading of drug relative to excipient is not viable for low volume depot injections, especially given the potency of FTC. Therefore, we studied the impact of increasing drug content on the formation of SSPNs, targeting the highest possible reproducible output; given the success of 5'-carbonate FTC carbamates bearing long alkyl chains at low drug loadings, this study was conducted using compounds **4**, **6**, and **8** (C$_4$, C$_6$ and C$_8$ analogues). A fivefold increase in drug loading (50 wt% of the SSPNs relative to excipients) yielded 36 hits, with only 5

of these derived from **4**, but 14 and 17 hits from **6** and **8**, respectively (Fig. 4). Further significant increase in the drug loading to a target 70 wt% resulted in just 4 hits from across **4** and **8** and no hits from **6** (Fig. 4). This highlights the complexity of predicting the outcomes of ETFD screening to form SDNs and SSPNs. Accelerated screening with low quantities of drug compound (10 wt%) was used to provide clear guidance for options containing higher drug loadings as these materials consume considerable amounts of prodrug compounds; options loaded at >50 wt% were subject to multiple repeats to establish robustness and reproducibility (Supplementary Table 3). Importantly, no degradation of the prodrugs was observed as a consequence of the ETFD process, as confirmed through HPLC analysis of fully dissolved formulations (Supplementary Figure 23); the solid high-loading SSPN formulations produced

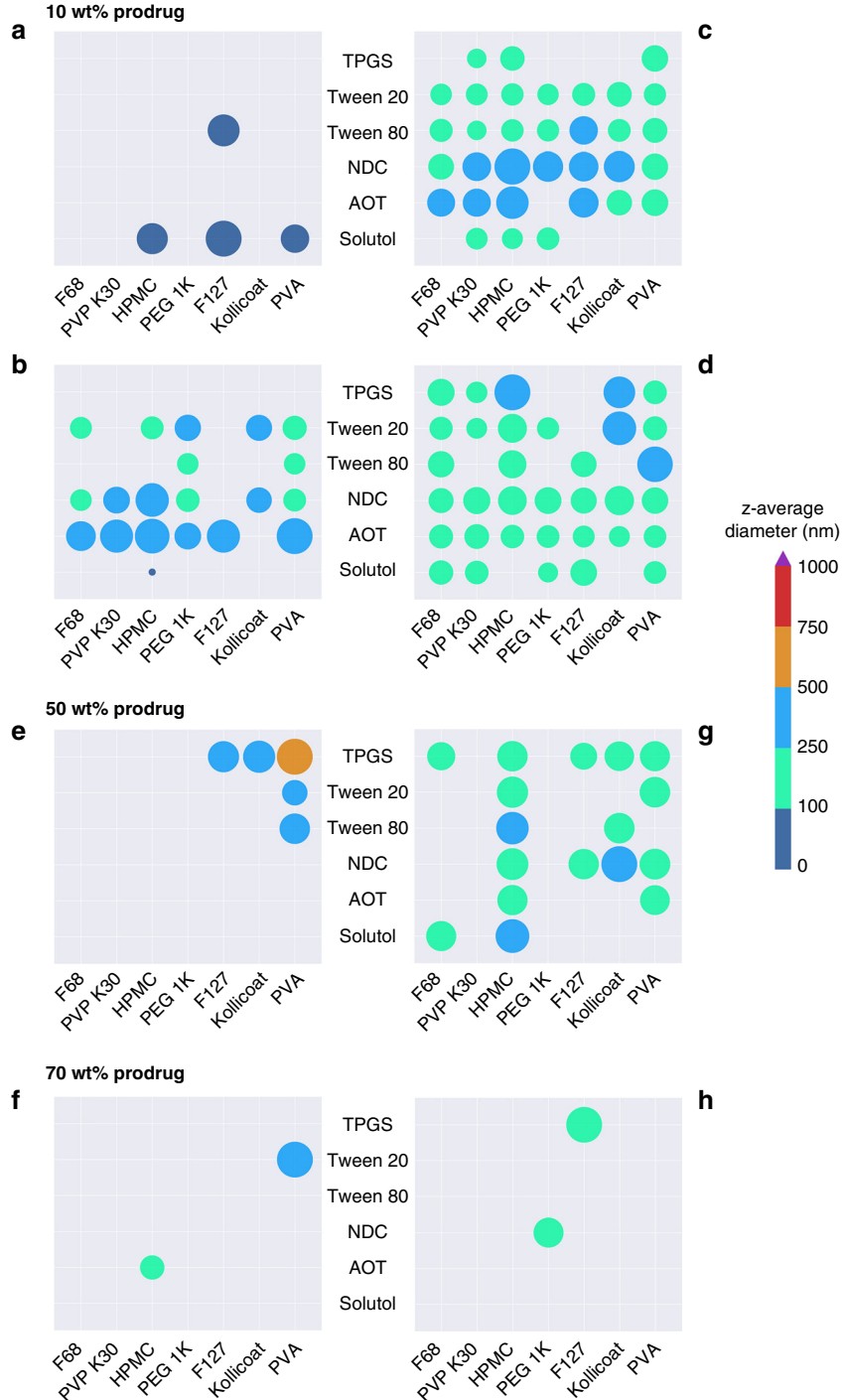

**Fig. 4** Overview of hits mapped onto binary excipient combinations. **a–d** Hits identified for prodrugs **2**, **4**, **6**, and **8** respectively at 10 wt% drug loading relative to excipients, **e**, **g** impact on number of hits and z-average diameter after increased prodrug loading to 50 wt% for **4** and **8** respectively, and **f**, **h** impact on number of hits and z-average diameter after increased prodrug loading to 70 wt% for **4** and **8** respectively. Each colour represents a size range (as indicated by the legend) and circle diameter represents whether the z-average diameter value resides at the lower or higher end of the range

here were highly robust and were able to maintain their dispersibility over several months and after long-distance transatlantic transportation.

**Pharmacological evaluation of SSPN candidate potential.** In vivo drug release from depot injections is difficult to predict due to the range of competing biological processes that occur after administration. Physiologically based pharmacokinetic modelling

has emerged as a valuable in silico tool to predict in vivo pharmacokinetics based on in vitro analysis. At this early stage of pre-clinical development, in vitro to in vivo extrapolation (IVIVE) can aid in assessing the potential for prodrug SSPNs to yield sustained in vivo release of FTC in humans and overcome shortcomings that emerge from reliance on animal models alone; a particular issue within long-acting drug release that is likely to be exacerbated by species differences in prodrug activation. A seven compartment IVIVE model was generated to simulate

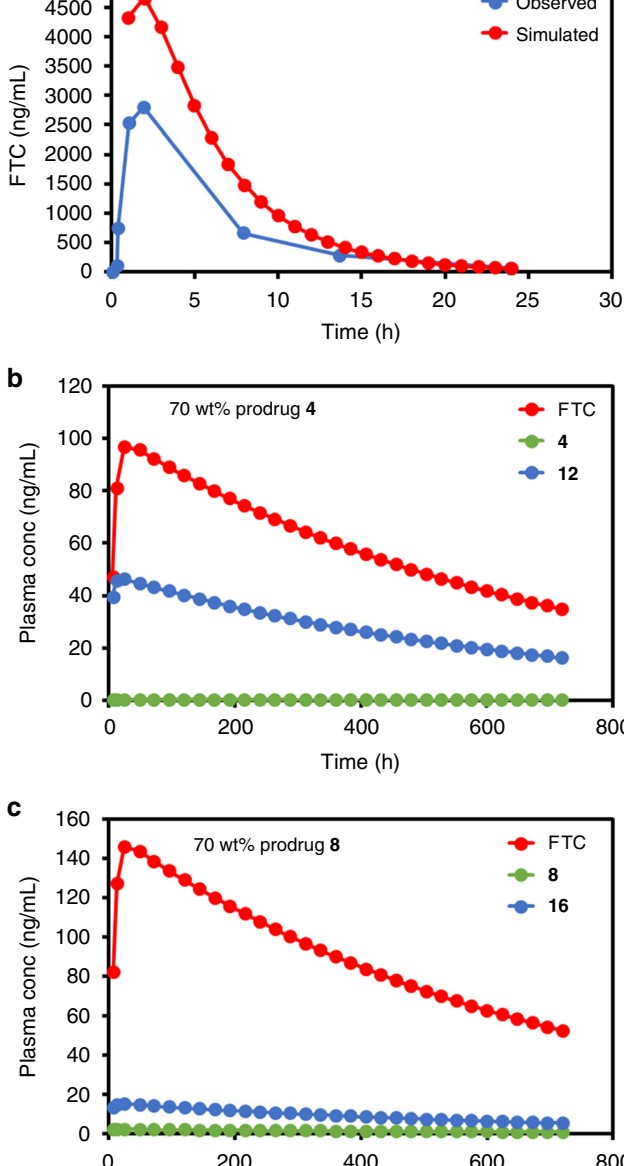

**Fig. 5** Simulated plasma concentrations and pharmacokinetics (PK) parameters following: **a** single dose oral administration of 400 mg emtricitabine (FTC), **b** intramuscular (IM) injection of semi-solid prodrug nanoparticle (SSPN) prodrug **4** (including activation to prodrug **12** and finally FTC), and **c** IM injection of SSPN containing prodrug **8** (including activation to prodrug **16** and finally FTC). Values represent mean (oral data ± minimum and maximum all other data ± standard deviation)

systemic exposure to FTC prodrugs (Supplementary Figure 45). In vitro data describing the metabolic prodrug activation to parent FTC were incorporated to simulate the human pharmacokinetics (PK) of FTC (Supplementary Table 4). The model was qualified by using a simulated daily oral dose of 400 mg of FTC and compared to available human clinical data (Supplementary Table 5)[18]. The simulated data showed acceptable deviation from the observed clinical data[19]. The $C_{max}$, $C_{min}$ and area under the curve varied by +62%, −10%, and +76%, respectively (Fig. 5a). The oral data demonstrated validation of the model within acceptable limits, with simulated data falling within twofold difference of the observed patient PK data[20].

At this stage in the development of prodrug SSPNs, there are no clinical data available for validation of the simulated prodrug PK. The qualified model was then used to simulate the human PK following a single IM injection of 2 g SSPN formulation (70 wt% prodrug) with either **4** or **8** over 28 days (Supplementary Tables 4–7). The maximum volume that can be administered intramuscularly is 6 mL[21,22], and a depot compartment to reflect this was created within the model. In order to achieve this, a formulation containing 333 mg/mL of prodrug would be required, which is comparable to other antiretroviral LA formulations such as rilpivirine[8]. The simulated human PK profiles for both **4** and **8** (Fig. 5b, c) demonstrated rapid conversion from the carbonate/carbamate to the intermediate carbamates **12** and **16** respectively (Fig. 2a for prodrug activation steps), $C_{max}$ (**4** = 0.5 ± 1.5 ng/mL, **8** = 2.2 ± 4.6 ng/mL) and $C_{min}$ (**4** = 0.2 ± 0.5 ng/mL, **8** = 0.8 ± 1.6 ng/mL).

The carbamate intermediate reached higher simulated concentrations but remained only a small contribution to total drug released, $C_{max}$ (**12** = 46.5 ± 11.9 ng/mL, **16** = 15.3 ± 7.4 ng/mL) and $C_{min}$ (**12** = 16.3 ± 4.2 ng/mL, **16** = 5.4 ± 2.7 ng/mL). Simulated FTC human PK demonstrated an early burst effect with $C_{max}$ reached at 24 h for **4** (97.0 ± 21.2 ng/mL) and **8** (146.4 ± 19.4 ng/mL) respectively. Following $C_{max}$, simulated FTC plasma concentrations demonstrated steady release over the 28 days reaching a $C_{min}$ of 34.9 (±7.7) ng/mL and 52.4 (± 6.8) ng/mL for depots containing SSPNs of prodrugs **4** and **8** respectively. Simulated FTC plasma concentrations fell below the concentration inhibiting 90% of viral replication ($IC_{90}$, 50 ng/mL) at 20 and 28 days for SSPNs of prodrugs **4** and **8**, respectively[23]. The intracellular concentrations of the active anabolite of FTC, FTC triphosphate (FTC-TP), were also simulated as previously described[23,24]. Simulated intracellular FTC-TP (Fig. 5b, c) $C_{max}$ were 618.8 (±302.5) fmol per $10^6$ cells and 921.2 (±391.4) fmol per $10^6$ cells. $C_{min}$ at day 28 maintained simulated intracellular concentrations above the intracellular half-maximal inhibitory concentration ($IC_{50}$; 150 fmol per $10^6$ cells), 433.1 (±211.1) fmol per $10^6$ cells, and 644.5 (±270.4) fmol per $10^6$ cells[25]. The simulated data indicate that SSPNs of prodrugs **4** and **8** have the potential to sustain concentrations of FTC above the intracellular $IC_{50}$ for 28 days following a single 2 g IM injection; in particular, the plasma concentrations of FTC derived from **8** are maintained above $IC_{90}$ values while intracellular concentrations stay above the $IC_{50}$. The simulations presented here were generated using input data determined experimentally, from literature or predicted parameters and validated against clinical pharmacokinetic data. While this modelling fits with convention, there was an apparent overestimation of plasma concentrations. Therefore, additional modelling was conducted whereby key model parameters such as plasma clearance were modified to force the model to fit the clinical data, and SSPNs of prodrugs **4** and **8** were again shown to have the potential to act as a viable long-acting depot injection (Supplementary Methods and Supplementary Tables 8–11).

## Discussion

The formation of a clinically relevant and viable long-acting injectable depot using prodrug strategies requires the balancing of a number of disparate factors including: ease of prodrug synthesis, appropriate activation kinetics, the ability to create an injectable formulation, and the maintenance of human in vivo pharmacokinetics that provide the correct exposure of the parent drug. Here we have shown that the iterative integration of prodrug design and synthesis with semi-solid nanoparticle formation and IVIVE predictive modelling can rapidly lead to the identification of lead candidates for progression into pre-clinical

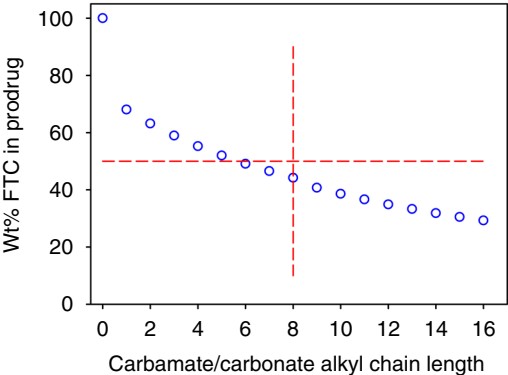

**Fig. 6** Reduction of parent emtrictabin (FTC) content with increasing alkyl chain length for carbonate/carbamate prodrugs. As a guide, a 50% mass penalty is highlighted by the horizontal red lines and compounds studied here only utilised chain lengths up to 8 carbon atoms (vertical red line)

evaluation; in programmes where multiple options exist, the use of IVIVE to estimate behaviour of the SSPN depot in human subjects to highlight candidates with clinical relevance also conforms strongly with the principles of the National Centre for the Replacement, Refinement and Reduction of Animals in Research (often referred to as the NC3Rs), where the use of technological developments to minimise animal experimentation is strongly recommended. From the 16 prodrug candidates and hundreds of possible SSPN formulations, the constraints imposed by the chosen manufacturing approach were able to rapidly reduce the candidates and highlight fully masked carbamate/carbonate prodrugs as having ideal properties for the ETFD approach to generate FTC prodrug SSPNs. The ETFD approach also allowed the relationship between prodrug loading in the formulation and alkyl chain length of the prodrug moieties to be elucidated. This context is particularly important when considering the mass penalty of masking the parent drug functional groups to generate a water-insoluble prodrug (Fig. 6).

Clearly, for low molecular weight drugs (FTC MW = 247.25 g mol$^{-1}$), the necessary addition of masking groups will reduce the dose of parent drug within the injection volume and depot site; indeed, if alkyl chain lengths of up to 16 carbons were utilised for both carbamate and carbonate masking groups, parent drug content would drop to <30 wt% of the prodrug mass. ETFD allows for high drug loading relative to excipients (up to 70 wt%), therefore, mitigating this penalty to some degree; IVIVE predictions allow confidence for progression to in vivo evaluations, as benefits in humans have been predicted. These are the next steps for our candidate materials; the combined approach presented here minimises animal testing in the search for candidates with both clinical and manufacturing relevance. While the simulated PK of SSPNs are highly encouraging, the simulated doses are high. Intramuscular injection volumes as high as 6 mL have been administered previously, but smaller volumes would be preferable particularly when repeated administration is required. Recently, co-administration of hyaluronidase has been used as a strategy to reduce patient discomfort and enable higher volume subcutaneous injections[26]. Hyaluronidase acts via hydrolysis of hyaluronan, a key component of the extracellular matrix, resulting in increased membrane permeability[27]. This approach has been used to increase the volume for both subcutaneous and intramuscular injections of anaesthetic agents, and subcutaneous administration of immunoglobulins.

Current treatment strategies focus on treating HIV via a combination of antiretrovirals and the EFTD technology is readily compatible with generating combination nanoparticles

comprised of more than one agent. This has previously been empirically demonstrated with lopinavir/ritonavir co-formulation and the presence of two agents within individual nanoparticles has been explicitly demonstrated using fluorescence resonance energy transfer dyes[12,28]. Future work will assess the potential to generate formulations containing multiple antiretroviral drugs.

The SSPN formulations are predicted to deliver FTC at an appropriate plasma concentration to match dosing regimens estimated for the clinically advanced cabotegravir and rilpivirine depot candidates. This offers the potential for water-soluble backbone NRTI delivery to supplement these industrial developments and increase the options for HIV patients globally; the combination of this SSPN LA option with the individual cabotegravir or rilpivirine long-acting injectable candidates would immediately double the long-acting injectable options available clinically. The potential for long-acting NRTI delivery, through the alteration of the parent drugs towards hydrophobic and poorly water-soluble prodrugs, opens considerable scope for future therapy development for various diseases, especially as low melting point materials have been highlighted as viable development candidates as SSPNs. Our ongoing studies aim to establish in vivo safety, pharmacokinetics, and efficacy to support progression towards human evaluation and create opportunities for additional patient benefits such as improved lifestyles, reduced pill burdens, and long-term stability of adherence.

## Methods

**General**. Unless otherwise noted, all reagents were obtained from commercial suppliers and used without further purification. Dichloromethane (DCM) was distilled after drying over CaH$_2$. Reaction yields refer to the purified products. Compound purification was carried out on a Biotage Isolera One flash chromatography system with indicated solvent mixtures and gradients. Elution was monitored by ultraviolet detection. Thin layer chromatography (TLC) was performed using 250 μm w/h F254 plates. $^1$H and $^{13}$C nuclear magnetic resonance (NMR) spectra were acquired on a Bruker Avance III 500 spectrometer operating at 500 MHz for $^1$H and 126 MHz for $^{13}$C. Chemical shift values are reported as δ (ppm) relative to CHCl$_3$ at δ 7.27 ppm, MeOH at δ 3.31 ppm, and dimethyl sulfoxide (DMSO) at δ 2.50 ppm for $^1$H NMR. Mass spectrometry analysis was performed on a Thermo Q-Exactive (ESI ionisation with orbitrap mass analyser). The purity of synthesised compounds is≥95% as analysed by HPLC (Beckman Gold Nouveau System Gold) on a C$_{18}$ column (Grace Altima, 3 μm C$_{18}$ analytical Rocket® column, 53 mm × 7 mm) using the following method: 0% to 100% B over 10 min at a flow rate of 3 mL min$^{-1}$ (solvent A: Et$_3$NHOAc (50 mM, pH 8), solvent B: acetonitrile). Unless otherwise noted, all HPLC analyses were performed using the following method: 0% to 100% B over 10 min at a flow rate of 3 mL min$^{-1}$ (solvent A: Et$_3$NHOAc (50 mM, pH 8), solvent B: acetonitrile). Calculated log*P* (clog*P*) values were derived from SMILES files using the Virtual Computational Chemistry Laboratory (vcclab.org).

**General synthesis of 5'-alkoxycarbonyl FTC carbamates (1–8)**. In a flame-dried 25 mL round-bottom flask, cooled under argon, FTC (1.0 eq., 0.5 M) was suspended in DCM. Pyridine (3.0 eq., 1.5 M) was then added to the flask, and the resulting mixture was cooled to 0 °C in an ice-water bath. The reaction was initiated by the dropwise addition of the alkyl chloroformate (2.1 eq., 1.05 M). The reaction mixture was allowed to warm to room temperature with stirring. The reaction was deemed complete after 3 h as monitored by TLC. Volatiles were removed from the reaction mixture under reduced pressure. The resulting residue was purified via silica flash chromatography (30% EtOAc in Hexanes for 5 min, then 30–100% EtOAc over 6 min, then 100% EtOAc for 3 min).

**Characterisation data for prodrugs 1–8**. Methyl (5-fluoro-1-((2R,5S)-2-(((methoxycarbonyl)oxy)methyl)-1,3-oxathiolan-5-yl)-2-oxo-1,2-dihydropyrimidin-4-yl)carbamate (1): 262 mg of clear, colourless waxy solid (68%). Mp: 97–103 °C. $^1$H NMR (500 MHz, CD$_3$OD) δ ppm 3.34 (br. s., 1 H) 3.63 (dd, *J* = 12.50, 5.27 Hz, 1 H) 3.77–3.84 (m, 6 H) 4.54–4.70 (m, 2 H) 5.49 (br. s., 1 H) 6.26 (br. s., 1 H) 8.33 (d, *J* = 6.45 Hz, 1 H). $^{13}$C NMR (126 MHz, CD$_3$OD) δ ppm 25.40 (s, 1 C) 39.21 (s, 1 C) 53.74 (s, 1 C) 55.96 (s, 1 C) 67.79 (s, 1 C) 86.22 (s, 1 C) 89.32 (s, 1 C) 129.77 (d, *J* = 32.40 Hz, 1 C) 138.58 (d, *J* = 240.00 Hz, 1 C) 154.02 (br. s., 1 C) 155.51 (d, *J* = 12.71 Hz, 1 C) 156.97 (s, 1 C). HRMS (ESI) m/z: calc'd 364.0609 [M + H]$^+$, 386.0429 [M + Na]$^+$; found 364.0601, 386.0420. RP-HPLC retention time (0% to 100% B over 5 min): 3.0 min.

Ethyl (1-((2R,5S)-2-(((ethoxycarbonyl)oxy)methyl)-1,3-oxathiolan-5-yl)-5-fluoro-2-oxo-1,2-dihydropyrimidin-4-yl)carbamate (2): 551 mg of clear, colourless waxy solid (89%). $^1$H NMR (500 MHz, CDCl$_3$) δ ppm 1.30–1.39 (m, 6 H) 3.21 (d, *J* = 10.38 Hz, 1 H) 3.53 (d, *J* = 6.92 Hz, 1 H) 4.25 (q, *J* = 7.07 Hz, 4 H) 4.57

(br. s., 2 H) 5.39 (t, $J$ = 3.22 Hz, 1 H) 6.31 (br. s., 1 H) 8.01 (br. s., 1 H) 12.08 (br. s., 1 H). $^{13}$C NMR (126 MHz, CDCl$_3$) δ ppm 13.91 (s, 1 C) 14.00 (s, 1 C) 38.08 (br. s., 1 C) 62.28 (br. s., 1 C) 64.64 (s, 1 C) 66.02 (br. s., 1 C) 84.12 (br. s., 1 C) 86.75 (br. s., 1 C) 124.56 (br. s., 1 C) 139.11 (d, $J$ = 240.00 Hz, 1 C) 145.92 (br. s., 1 C) 153.19 (d, $J$ = 18.17 Hz, 1 C) 154.48 (s, 1 C) 163.28 (s, 1 C). HRMS (ESI) m/z: calc'd 392.0922 [M + H]$^+$, 414.0742 [M + Na]$^+$; found 392.0912, 414.0732. RP-HPLC retention time (0% to 100% B over 5 min): 3.4 min.

Propyl (5-fluoro-2-oxo-1-((2R,5S)-2-(((propoxycarbonyl)oxy)methyl)-1,3-oxathiolan-5-yl)-1,2-dihydropyrimidin-4-yl)carbamate (3): 741 mg of clear, colourless waxy solid (88%). Mp: 102–107 °C. $^1$H NMR (500 MHz, CDCl$_3$) δ ppm 0.98 (dt, $J$ = 11.00, 7.47 Hz, 6 H) 1.66–1.81 (m, 4 H) 3.21 (d, $J$ = 10.22 Hz, 1 H) 3.54 (br. s., 1 H) 4.16 (t, $J$ = 6.45 Hz, 4 H) 4.57 (br. s., 2 H) 5.39 (br. s., 1 H) 6.31 (br. s., 1 H) 8.02 (br. s., 1 H) 12.10 (br. s., 1 H). $^{13}$C NMR (126 MHz, CDCl$_3$) δ ppm 10.05 (s, 1 C) 10.31 (br. s., 1 C) 21.85 (s, 1 C) 21.88 (s, 1 C) 38.24 (br. s., 1 C) 66.15 (br. s., 1 C) 68.07 (br. s., 1 C) 70.39 (br. s., 1 C) 84.22 (br. s., 1 C) 86.71 (br. s., 1 C) 124.52 (d, $J$ = 32.40 Hz, 1 C) 139.36 (d, $J$ = 240.00 Hz, 1 C) 146.06 (br. s., 1 C) 153.39 (d, $J$ = 16.00 Hz, 1 C) 154.77 (s, 1 C) 163.59 (br. s., 1 C). HRMS (ESI) m/z: calc'd 420.1235 [M + H]$^+$, 442.1055 [M + Na]$^+$; found 420.126, 442.1045. RP-HPLC retention time (0% to 100% B over 5 min): 3.7 min.

Butyl (1-((2R,5S)-2-(((butoxycarbonyl)oxy)methyl)-1,3-oxathiolan-5-yl)-5-fluoro-2-oxo-1,2-dihydropyrimidin-4-yl)carbamate (4): 901 mg of clear, colourless waxy solid (95%). $^1$H NMR (500 MHz, CDCl$_3$) δ ppm 0.94 (td, $J$ = 7.39, 2.20 Hz, 6 H) 1.41 (tq, $J$ = 14.72, 7.43 Hz, 4 H) 3.20 (d, $J$ = 9.59 Hz, 1 H) 3.53 (d, $J$ = 9.60 Hz, 1 H) 4.19 (t, $J$ = 6.68 Hz, 4 H) 4.56 (br. s., 2 H) 5.39 (t, $J$ = 3.22 Hz, 1 H) 6.30 (br. s., 1 H) 8.00 (br. s., 1 H) 12.09 (br. s., 1 H). $^{13}$C NMR (126 MHz, CDCl$_3$) δ ppm 13.59 (s, 1 C) 13.66 (s, 1 C) 18.78 (s, 1 C) 19.01 (s, 1 C) 30.47 (s, 1 C) 30.56 (s, 1 C) 38.21 (br. s., 1 C) 66.30 (br. s., 2 C) 68.73 (s, 1 C) 84.16 (br. s., 1 C) 86.75 (br. s., 1 C) 124.47 (d, $J$ = 38.00 Hz, 1 C) 139.41 (d, $J$ = 240.00 Hz, 1 C) 146.09 (br. s., 1 C) 153.32 (br. s., 1 C) 154.81 (br. s., 1 C) 163.64 (br. s., 1 C). HRMS (ESI) m/z: calc'd 448.1548 [M + H]$^+$, 470.1368 [M + Na]$^+$; found 448.1551, 470.1367. RP-HPLC retention time (0% to 100% B over 5 min): 4.2 min.

Pentyl (5-fluoro-2-oxo-1-((2R,5S)-2-((((pentyloxy)carbonyl)oxy)methyl)-1,3-oxathiolan-5-yl)-1,2-dihydropyrimidin-4-yl)carbamate (5): 1430 mg of clear, colourless waxy solid (90%). Mp: 103–106 °C. $^1$H NMR (500 MHz, CDCl$_3$) δ ppm 0.83–0.98 (m, 6 H) 1.29–1.44 (m, 8 H) 1.62–1.77 (m, 4 H) 3.20 (d, $J$ = 10.38 Hz, 1 H) 3.53 (d, $J$ = 7.39 Hz, 1 H) 4.15–4.29 (m, 4 H) 4.56 (br. s., 2 H) 5.39 (t, $J$ = 3.22 Hz, 1 H) 6.30 (br. s., 1 H) 7.99 (br. s., 1 H) 12.10 (br. s., 1 H). $^{13}$C NMR (126 MHz, CDCl$_3$) δ ppm 13.80 (s, 1 C) 13.83 (s, 1 C) 22.10 (s, 1 C) 22.20 (s, 1 C) 27.55 (s, 1 C) 27.81 (s, 1 C) 28.06 (s, 1 C) 28.16 (s, 1 C) 38.06 (br. s., 1 C) 66.19 (br. s., 1 C) 66.56 (br. s., 1 C) 68.88 (br. s., 1 C) 84.00 (br. s., 1 C) 86.70 (br. s., 1 C) 124.47 (d, $J$ = 32.40 Hz, 1 C) 139.31 (d, $J$ = 240.00 Hz, 1 C) 145.97 (br. s., 1 C) 153.19 (br. s., 1 C) 154.71 (s, 1 C) 163.57 (br. s., 1 C). HRMS (ESI) m/z: calc'd 476.1861 [M + H]$^+$, 498.1681 [M + Na]$^+$; found 476.1853, 498.1669. RP-HPLC retention time (0% to 100% B over 5 min): 4.6 min.

Hexyl (5-fluoro-1-((2R,5S)-2-((((hexyloxy)carbonyl)oxy)methyl)-1,3-oxathiolan-5-yl)-2-oxo-1,2-dihydropyrimidin-4-yl)carbamate (6): 904 mg of clear, colourless waxy solid (95%). Mp: 110 °C. $^1$H NMR (500 MHz, CDCl$_3$) δ ppm 0.78–0.97 (m, 6 H) 1.21–1.45 (m, 12 H) 1.60–1.74 (m, 4 H) 3.19 (d, $J$ = 9.90 Hz, 1 H) 3.53 (br. s., 1 H) 4.18 (t, $J$ = 6.68 Hz, 4 H) 4.47 - 4.69 (m, 2 H) 5.38 (t, $J$ = 3.22 Hz, 1 H) 6.30 (br. s., 1 H) 7.99 (br. s., 1 H) 12.09 (br. s., 1 H). $^{13}$C NMR (126 MHz, CDCl$_3$) δ ppm 13.94 (s, 1 C) 13.98 (s, 1 C) 22.46 (s, 1 C) 22.50 (s, 1 C) 25.19 (s, 1 C) 25.47 (br. s., 1 C) 28.42 (s, 1 C) 28.51 (s, 1 C) 31.28 (s, 1 C) 31.40 (s, 1 C) 38.13 (br. s., 1 C) 66.28 (br. s., 1 C) 66.64 (br. s., 1 C) 69.01 (s, 1 C) 84.10 (br. s., 1 C) 86.75 (br. s., 1 C) 124.43 (d, $J$ = 32.40 Hz, 1 C) 139.42 (d, $J$ = 240.00 Hz, 1 C) 146.06 (br. s., 1 C) 153.30 (br. s., 1 C) 154.79 (s, 1 C) 163.62 (br. s., 1 C). HRMS (ESI) m/z: calc'd 504.1274 [M + H]$^+$, 526.1994 [M + Na]$^+$; found 504.2178, 526.1996. RP-HPLC retention time (0% to 100% B over 5 min): 5.1 min.

Heptyl (5-fluoro-1-((2R,5S)-2-((((heptyloxy)carbonyl)oxy)methyl)-1,3-oxathiolan-5-yl)-2-oxo-1,2-dihydropyrimidin-4-yl)carbamate (7): 1062 mg of clear, colourless waxy solid (99%). $^1$H NMR (500 MHz, CDCl$_3$) δ ppm 0.82–0.95 (m, 6 H) 1.18–1.43 (m, 16 H) 1.66–1.75 (m, 4 H) 3.20 (br. s., 1 H) 3.53 (br. s., 1 H) 4.16 (t, $J$ = 6.30 Hz, 4 H) 4.56 (br. s., 2 H) 5.38 (t, $J$ = 3.30 Hz, 1 H) 6.30 (br. s., 1 H) 7.99 (br. s., 1 H) 12.09 (br. s., 1 H). $^{13}$C NMR (126 MHz, CDCl$_3$) δ ppm 14.02 (s, 2 C) 22.51 (s, 1 C) 22.53 (s, 1 C) 25.47 (s, 1 C) 25.74 (br. s., 1 C) 28.45 (s, 1 C) 28.54 (s, 1 C) 28.78 (s, 1 C) 28.88 (s, 1 C) 31.63 (s, 1 C) 31.66 (s, 1 C) 38.14 (br. s., 1 C) 66.27 (br. s., 1 C) 66.62 (br. s., 1 C) 69.01 (s, 1 C) 84.10 (br. s., 1 C) 86.73 (br. s., 1 C) 124.47 (d, $J$ = 32.40 Hz, 1 C) 139.43 (d, $J$ = 240.00 Hz, 1 C) 146.03 (br. s., 1 C) 153.42 (br. s., 1 C) 154.78 (s, 1 C) 163.59 (br. s., 1 C). HRMS (ESI) m/z: calc'd 532.2487 [M + H]$^+$, 554.2307 [M + Na]$^+$; found 532.2480, 554.2296. RP-HPLC retention time (0% to 100% B over 5 min): 5.4 min.

Octyl (5-fluoro-1-((2R,5S)-2-((((octyloxy)carbonyl)oxy)methyl)-1,3-oxathiolan-5-yl)-2-oxo-1,2-dihydropyrimidin-4-yl)carbamate (8): 805 mg of clear, colourless waxy solid (89%). Mp: 216–218 °C. $^1$H NMR (500 MHz, CDCl$_3$) δ ppm 0.88 (t, $J$ = 1.00 Hz, 6 H) 1.21–1.42 (m, 20 H) 1.66–1.73 (m, 4 H) 3.19 (d, $J$ = 10.38 Hz, 1 H) 3.53 (d, $J$ = 6.60 Hz, 1 H) 4.18 (t, $J$ = 6.40 Hz, 4 H) 4.56 (br. s., 2 H) 5.39 (t, $J$ = 3.22 Hz, 1 H) 6.30 (br. s., 1 H) 7.99 (br. s., 1 H) 12.10 (br. s., 1 H). $^{13}$C NMR (126 MHz, CDCl$_3$) δ ppm 14.06 (s, 2 C) 22.60 (s, 1 C) 25.53 (s, 1 C) 25.81 (br. s., 1 C) 28.47 (s, 1 C) 28.56 (s, 1 C) 29.09 (s, 1 C) 29.11 (s, 1 C) 29.14 (br. s., 1 C) 29.19 (s, 1 C) 31.73 (s, 1 C) 31.75 (s, 1 C) 38.15 (br. s., 1 C) 66.29 (br. s., 1 C) 66.65 (br. s., 1 C) 68.36 (s, 1 C) 69.04 (s, 1 C) 84.11 (br. s., 1 C) 86.76 (br. s., 1 C) 124.42 (d, $J$ = 32.40 Hz, 1 C) 139.39 (d, $J$ = 240.00 Hz, 1 C) 146.06 (br. s., 1 C) 153.30 (s, 1 C)

154.81 (s, 1 C) 163.62 (br. s., 1 C). HRMS (ESI) m/z: calc'd 560.2800 [M + H]$^+$, 582.2620 [M + Na]$^+$; found 560.2803, 582.2617. RP-HPLC retention time (0% to 100% B over 5 min): 5.9 min.

**General synthesis of FTC carbamates (9-16)**. In a 20 mL vial, the 5′-alkoxycarbonyl FTC carbamate (1–8, 1.0 eq., 0.5 M) was dissolved in tetrahydrofuran. Lithium hydroxide (5 eq., 2.5 M) was then added to the vial. Water (~20 drops) was added to the mixture dropwise to enhance solubility of the mixture. The reaction mixture was stirred at room temperature and monitored by TLC. After stirring for 18 h, the reaction was found to be complete by TLC. Volatiles were removed from the reaction mixture under reduced pressure. The resulting residue was purified via silica flash chromatography (3% MeOH in DCM for 8 min, then 3–10% MeOH over 5 min, then 10% MeOH for 3 min).

**Characterisation data for prodrugs 9-16**. Methyl (5-fluoro-1-((2R,5S)-2-(hydroxymethyl)-1,3-oxathiolan-5-yl)-2-oxo-1,2-dihydropyrimidin-4-yl)carbamate (9): 24 mg of white solid (58%). Mp: 85–89 °C. $^1$H NMR (500 MHz, CD$_3$OD) δ ppm 3.27–3.30 (m, 1 H) 3.61 (dd, $J$ = 12.58, 5.34 Hz, 1 H) 3.80 (s, 3 H) 3.90 (dd, $J$ = 12.81, 3.07 Hz, 1 H) 4.08 (dd, $J$ = 12.80, 2.80 Hz, 1 H) 5.33 (t, $J$ = 2.91 Hz, 1 H) 6.22 - 6.27 (m, 1 H) 8.79 (d, $J$ = 6.76 Hz, 1 H). $^{13}$C NMR (126 MHz, CD$_3$OD) δ ppm 39.72 (s, 1 C) 53.66 (s, 1 C) 62.77 (s, 1 C) 89.30 (s, 1 C) 90.35 (s, 1 C) 130.76 (d, $J$ = 32.40 Hz, 1 C) 136.47 (s, 1 C) 138.64 (d, $J$ = 240.00 Hz, 1 C) 154.47 (br. s., 1 C) 155.79 (d, $J$ = 13.62 Hz, 1 C). HRMS (ESI) m/z: calc'd 306.0554 [M + H]$^+$, 328.0374 [M + Na]$^+$; found 306.0551, 328.0369. RP-HPLC retention time (0% to 100% B over 5 min): 2.7 min.

Ethyl (5-fluoro-1-((2R,5S)-2-(hydroxymethyl)-1,3-oxathiolan-5-yl)-2-oxo-1,2-dihydropyrimidin-4-yl)carbamate (10): 161 mg of white solid (59%). $^1$H NMR (500 MHz, CD$_3$OD) δ ppm 1.32 (t, $J$ = 7.15 Hz, 3 H) 3.29 (dd, $J$ = 12.58, 2.83 Hz, 1 H) 3.60 (dd, $J$ = 12.42, 5.34 Hz, 1 H) 3.90 (dd, $J$ = 12.73, 3.14 Hz, 1 H) 4.07 (dd, $J$ = 12.81, 2.91 Hz, 1 H) 4.24 (q, $J$ = 7.07 Hz, 2 H) 5.32 (t, $J$ = 2.99 Hz, 1 H) 6.24 (ddd, $J$ = 4.95, 2.91, 1.57 Hz, 1 H) 8.70 (d, $J$ = 6.76 Hz, 1 H). $^{13}$C NMR (126 MHz, CD$_3$OD) δ ppm 14.72 (s, 1 C) 39.79 (br. s., 1 C) 62.74 (s, 1 C) 63.46 (s, 1 C) 89.33 (br. s., 1 C) 90.45 (br. s., 1 C) 131.37 (br. s., 1 C) 138.42 (d, $J$ = 240.00 Hz, 1 C) 152.63 (br. s., 1 C) 155.13 (br. s., 1 C) 155.70 (br. s., 1 C). HRMS (ESI) m/z: calc'd 320.0711 [M + H]$^+$, 342.0530 [M + Na]$^+$; found 320.0707, 342.0526. RP-HPLC retention time (0% to 100% B over 5 min): 2.9 min.

Propyl (5-fluoro-1-((2R,5S)-2-(hydroxymethyl)-1,3-oxathiolan-5-yl)-2-oxo-1,2-dihydropyrimidin-4-yl)carbamate (11): 74 mg of white solid (46%). Mp: 82–86 °C. $^1$H NMR (500 MHz, CD$_3$OD) δ ppm 0.99 (t, $J$ = 7.47 Hz, 3 H) 1.72 (sxt, $J$ = 7.14 Hz, 2 H) 3.26–3.30 (m, 1 H) 3.61 (dd, $J$ = 12.58, 5.19 Hz, 1 H) 3.90 (dd, $J$ = 12.89, 2.99 Hz, 1 H) 4.08 (dd, $J$ = 12.81, 2.59 Hz, 1 H) 4.16 (t, $J$ = 6.68 Hz, 2 H) 5.33 (t, $J$ = 2.91 Hz, 1 H) 6.24 (dt, $J$ = 3.07, 1.77 Hz, 1 H) 8.79 (br. s., 1 H). $^{13}$C NMR (126 MHz, CD$_3$OD) δ ppm 10.73 (s, 1 C) 23.20 (s, 1 C) 39.71 (s, 1 C) 62.75 (s, 1 C) 68.97 (s, 1 C) 89.26 (s, 1 C) 90.34 (s, 1 C) 130.95 (d, $J$ = 32.40 Hz, 1 C) 138.53 (d, $J$ = 240.00 Hz, 1 C) 153.09 (br. s., 1 C) 154.59 (br. s., 1 C) 155.60 (d, $J$ = 12.72 Hz, 1 C). HRMS (ESI) m/z: calc'd 334.0867 [M + H]$^+$, 356.0687 [M + Na]$^+$; found 334.0864, 356.0683. RP-HPLC retention time (0% to 100% B over 5 min): 3.0 min.

Butyl (5-fluoro-1-((2R,5S)-2-(hydroxymethyl)-1,3-oxathiolan-5-yl)-2-oxo-1,2-dihydropyrimidin-4-yl)carbamate (12): 134 mg of white solid (58%). Mp: 47 °C. $^1$H NMR (500 MHz, CDCl$_3$) ppm 0.89 (t, $J$ = 7.23 Hz, 3 H) 1.28–1.46 (m, 2 H) 1.53–1.72 (m, 2 H) 3.22 (d, $J$ = 12.26 Hz, 1 H) 3.48 (d, $J$ = 12.30 Hz, 1 H) 3.97 (d, $J$ = 11.00 Hz, 1 H) 4.06–4.26 (m, 3 H) 5.27 (br. s., 1 H) 6.20 (br. s., 1 H) 8.55 (br. s., 1 H) 11.97 (br. s., 1 H). $^{13}$C NMR (126 MHz, CDCl$_3$) δ ppm 13.58 (s, 1 C) 18.89 (s, 1 C) 30.48 (s, 1 C) 38.78 (br. s., 1 C) 61.95 (br. s., 1 C) 66.32 (br. s., 1 C) 87.19 (br. s., 1 C) 88.67 (br. s., 1 C) 126.42 (br. s., 1 C) 138.36 (d, $J$ = 240.00 Hz, 1 C) 145.94 (s, 1 C) 153.40 (br. s., 1 C) 163.19 (br. s., 1 C). HRMS (ESI) m/z: calc'd 348.1024 [M + H]$^+$, 370.0843 [M + Na]$^+$; found 348.1017, 370.0837. RP-HPLC retention time (0% to 100% B over 5 min): 3.2 min.

Pentyl (5-fluoro-1-((2R,5S)-2-(hydroxymethyl)-1,3-oxathiolan-5-yl)-2-oxo-1,2-dihydropyrimidin-4-yl)carbamate (13): 750 mg of white solid (98%). Mp: 44–47 °C $^1$H NMR (500 MHz, CDCl$_3$) δ ppm 0.86–0.92 (m, 3 H) 1.26–1.42 (m, 4 H) 1.68 (quin, $J$ = 7.03 Hz, 2 H) 3.24 (dd, $J$ = 12.58, 2.83 Hz, 1 H) 3.51 (br. s., 1 H) 3.91–4.06 (m, 1 H) 4.09–4.26 (m, 3 H) 5.30 (t, $J$ = 2.52 Hz, 1 H) 6.24 (ddd, $J$ = 4.99, 3.10, 1.34 Hz, 1 H) 8.52 (br. s., 1 H). $^{13}$C NMR (126 MHz, CDCl$_3$) δ ppm 13.98 (s, 1 C) 22.33 (s, 1 C) 27.92 (s, 1 C) 28.31 (s, 1 C) 38.93 (br. s., 1 C) 62.12 (br. s., 1 C) 66.73 (br. s., 1 C) 87.38 (br. s., 1 C) 88.83 (br. s., 1 C) 126.64 (br. s., 1 C) 138.67 (d, $J$ = 240.00 Hz, 1 C) 146.29 (br. s., 1 C) 153.61 (br. s., 1 C) 163.53 (br. s., 1 C). HRMS (ESI) m/z: calc'd 362.1180 [M + H]$^+$, 384.1000 [M + Na]$^+$; found 362.1174, 384.0993. RP-HPLC retention time (0% to 100% B over 5 min): 3.5 min.

Hexyl (5-fluoro-1-((2R,5S)-2-(hydroxymethyl)-1,3-oxathiolan-5-yl)-2-oxo-1,2-dihydropyrimidin-4-yl)carbamate (14): 87 mg of white solid (64%). Mp: 46–49 °C. $^1$H NMR (500 MHz, CDCl$_3$) δ ppm 0.90 (t, $J$ = 7.00 Hz, 3 H) 1.26–1.47 (m, 6 H) 1.70 (quin, $J$ = 7.00 Hz, 2 H) 3.21 (d, $J$ = 11.16 Hz, 1 H) 3.52 (d, $J$ = 8.02 Hz, 1 H) 3.97 (d, $J$ = 12.10 Hz, 1 H) 4.17 (br. s., 3 H) 5.33 (t, $J$ = 3.85 Hz, 1 H) 6.30 (br. s., 1 H) 8.25 (br. s., 1 H) 12.11 (br. s., 1 H). $^{13}$C NMR (126 MHz, CDCl$_3$) δ ppm 13.89 (s, 1 C) 22.40 (s, 1 C) 25.33 (s, 1 C) 28.44 (s, 1 C) 31.29 (s, 1 C) 38.76 (br. s., 1 C) 61.97 (br. s., 1 C) 66.59 (s, 1 C) 87.22 (br. s., 1 C) 88.63 (br. s., 1 C) 126.87 (s, 1 C) 138.38 (d, $J$ = 240.00 Hz, 1 C) 146.09 (s, 1 C) 153.52 (d, $J$ = 14.53 Hz, 1 C) 163.08 (s, 1 C). HRMS (ESI) m/z: calc'd 376.1337 [M + H]$^+$, 398.1156 [M + Na]$^+$; found 376.1332, 398.1151. RP-HPLC retention time (0% to 100% B over 5 min): 3.5 min.

Heptyl (5-fluoro-1-((2R,5S)-2-(hydroxymethyl)-1,3-oxathiolan-5-yl)-2-oxo-1,2-dihydropyrimidin-4-yl)carbamate (15): 230 mg of white solid (67%). Mp: 46 °C. $^1$H NMR (500 MHz, CDCl$_3$) δ ppm 0.85 (t, $J = 7.00$ Hz, 3 H) 1.16–1.41 (m, 8 H) 1.65 (quin, $J = 6.50$ Hz, 2 H) 3.23 (dd, $J = 12.65, 2.75$ Hz, 1 H) 3.50 (d, $J = 7.70$ Hz, 1 H) 3.98 (dd, $J = 12.89, 2.83$ Hz, 1 H) 4.08 - 4.24 (m, 3 H) 5.28 (t, $J = 2.59$ Hz, 1 H) 6.17 - 6.29 (m, 1 H) 8.55 (br. s., 1 H) 12.07 (br. s., 1 H). $^{13}$C NMR (126 MHz, CDCl$_3$) δ ppm 13.94 (s, 1 C) 22.43 (s, 1 C) 25.62 (s, 1 C) 28.48 (s, 1 C) 28.78 (s, 1 C) 31.57 (s, 1 C) 38.76 (br. s., 1 C) 61.98 (br. s., 1 C) 66.59 (br. s., 1 C) 87.18 (br. s., 1 C) 88.66 (br. s., 1 C) 126.26 (br. s., 1 C) 138.67 (d, $J = 240.00$ Hz, 1 C) 146.10 (s, 1 C) 153.55 (br. s., 1 C) 163.25 (s, 1 C). HRMS (ESI) m/z: calc'd 390.1493 [M + H]$^+$, 412.1313 [M + Na]$^+$; found 390.1488, 412.1307. RP-HPLC retention time (0% to 100% B over 5 min): 3.8 min.

Octyl (5-fluoro-1-((2R,5S)-2-(hydroxymethyl)-1,3-oxathiolan-5-yl)-2-oxo-1,2-dihydropyrimidin-4-yl)carbamate (16): 130 mg of white solid (81%). Mp: 41–44 °C. $^1$H NMR (500 MHz, CDCl$_3$) δ ppm 0.82 (t, $J = 6.60$ Hz, 3 H) 1.16–1.37 (m, 10 H) 1.56–1.69 (m, 2 H) 3.20 (dd, $J = 12.42, 1.73$ Hz, 1 H) 3.47 (d, $J = 8.17$ Hz, 1 H) 3.95 (dd, $J = 12.58, 1.57$ Hz, 1 H) 4.08–4.21 (m, 3 H) 5.26 (br. s., 1 H) 6.19 (br. s., 1 H) 8.55 (br. s., 1 H) 12.03 (br. s., 1 H). $^{13}$C NMR (126 MHz, CDCl$_3$) δ ppm 13.99 (s, 1 C) 22.53 (s, 1 C) 25.69 (s, 1 C) 28.50 (s, 1 C) 29.07 (s, 1 C) 29.11 (s, 1 C) 31.67 (s, 1 C) 38.73 (br. s., 1 C) 62.00 (br. s., 1 C) 66.62 (br. s., 1 C) 87.19 (br. s., 1 C) 88.58 (br. s., 1 C) 126.15 (br. s., 1 C) 138.65 (d, $J = 240.00$ Hz, 1 C) 145.92 (br. s., 1 C) 153.58 (br. s., 1 C) 163.22 (br. s., 1 C). HRMS (ESI) m/z: calc'd 404.1650 [M + H]$^+$, 426.1469 [M + Na]$^+$; found 404.1645, 426.1462. RP-HPLC retention time (0% to 100% B over 5 min): 4.0 min.

**Carbonate cleavage initial rate measurements (1–8).** Reaction mixtures containing mixed gender human skeletal muscle S9 (10.47 mg mL$^{-1}$, Bioreclamation) or phosphate buffer (0.1 M, pH 7.4) and pooled, mixed gender liver S9 (2.0 mg mL$^{-1}$, Xenotech) were pre-incubated at 37 °C for 5 min. Reactions were initiated by the addition of prodrugs (1 mM + 5% DMSO). After incubation at 37 °C, aliquots were taken at time points measuring initial rate (Muscle S9: **1–3**: 0.5, 10, 20, 30, 40, 50, and 60 min; **4**: 0.5, 20, 40, 60, 80, 100, and 120 min; **5–6**: 0.5, 30, 60, 90, 150, and 180 min; **7–8**: 0.5 min, 30 min, 1 h, 3 h, 4 h, 5 h) (Liver S9: **1–2**: 0.5, 1, 2, 3, 4, and 5 min; **3–6**: 0.5, 2, 4, 6, 8, and 10 min; **7–8**: 0.5, 5, 10, 15, 20, 25, 30, and 60 min). These aliquots were quenched in two volumes of ice-cold methanol. The quenched aliquots were then centrifuged at 16,873 × g for 5 min. The supernatant was diluted 10-fold into phosphate buffer (0.1 M, pH 7.4) and analysed by HPLC method described within Supplementary Information (monitoring starting material depletion at 305 nm).

**Carbonate cleavage half-life measurements (1–8).** Reaction mixtures containing pooled mixed gender human plasma (Bioreclamation) were pre-incubated at 37 °C for 5 min. Reactions were initiated by the addition of prodrugs **1–8** (1 mM + 5% DMSO). After incubation at 37 °C, aliquots were taken at the following time points: **1–3**: 0.5, 10, 20, 30, 40, 50, and 60 min; **4**: 0.5, 20, 40, 60, 80, 100, and 120 min; **5–6**: 0.5, 30, 60, 90, 120, 150, and 180 min; **7**: 0.5 min, 30 min, 1 h, 2 h, 3 h, 4 h, 5 h; **8**: 0.5, 20, 30, 50, and 90 min. These aliquots were quenched in two volumes of ice-cold methanol. The quenched aliquots were then centrifuged at 16,873 × g for 5 min. The supernatant was diluted 10-fold into phosphate buffer (0.1 M, pH 7.4) and analysed by HPLC method described within Supplementary Information (monitoring starting material depletion at 305 nm).

**General semi-solid prodrug nanoparticle methods.** For all wt% loadings of SSPNs, Ultra-Sonication was performed using a Covaris S220x sonicator, controlled by SonoLab 7.2 software. Photon Correlation Spectroscopy analysis was carried out using a Malvern ZetaSizer Nano S DLS machine. Freeze drying was performed using a VirTis BenchTop K freeze dryer, attached to a aerlikon TRIVAC D4B vacuum pump.

**ETFD synthesis of SSPNs at 10 wt% prodrug loading.** Into separate 14 mL glass sample vials, polymer and surfactants were weighed out and dissolved to a final concentration of 22.5 mg mL$^{-1}$ in distilled water. These solutions were left overnight on a rolling mixer to ensure thorough dissolution. Immediately before synthesis of SSPNs, prodrug was removed from the freezer and weighed out into a fresh 14 mL glass vial. Prodrug was dissolved to a final concentration of 10 mg mL$^{-1}$ in chloroform and left on a rolling mixer for 10 min to ensure thorough dissolution. Prodrug was not left rolling in solution for any excess time to prevent hydrolysis. For each SSPN sample, 133.4 μL of surfactant, 266.6 μL of polymer, and 100 μL of prodrug were added to a 4 mL glass sample vial. This was repeated for all 42 combinations of polymer and surfactant (7 × 6) and each prodrug. The final composition yielded SSPNs consisting of 1 mg prodrug (10 wt%), 3 mg surfactant (30 wt%), and 6 mg polymer (60 wt%). This 1:4 ratio of chloroform to water emulsion was sonicated for 15 s with the following protocol: 20% duty cycle; 250 intensity; 500 cycles/burst; frequency sweeping mode. This protocol gave an average output of 70 W. Samples were sonicated in a temperature-controlled water bath set to 4 °C. Immediately after sonication, emulsion samples were frozen in liquid nitrogen, prior to freeze drying using a VirTis BenchTop K freeze dryer (SP Scientific, Ipswich, UK) set to −100 °C and vacuum of <40 μbar. Sample remained in the freezer dryer for 48 h, after which they were stored in a desiccator at ambient temperature, prior to analysis.

## Data availability

All data generated during this study supporting its findings are available within the paper and the Supplementary Information. All data are available from the corresponding author upon reasonable request.

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

## Acknowledgements

This work was funded by the US National Institutes of Health (NIH) under grant number R01AI114405-01, and by a Ruth L. Kirschstein National Research Service Award (NRSA) pre-doctoral fellowship from the NIH (F31 AI129549, awarded to A.A.-k.). The authors also wish to acknowledge the Centre for Materials Discovery for access to equipment, Carley Heck and Namandje Bumpus for assistance obtaining analytical data for the prodrugs studied, Stephanie Henriquez for assistance obtaining standard curves and NMR characterisation data of prodrugs, and Nicolas Bruneau for assistance with the data visualisation script (PYTHON).

## Author contributions

C.F.M., C.F., M.S., A.O. and S.P.R. secured the funding, conceived the project, and supervised the research. J.J.H. conducted the emulsion-templated freeze drying screening, optimisation, and resulting SSPN physical characterisation. A.A.-k. and D.M. conducted prodrug synthesis, characterisation, and activation studies. P.C. conducted the IVIVE modelling. C.F.M., A.O. and S.P.R. wrote and edited the manuscript with contributions from P.C., A.A.-k. and J.J.H.

## Additional information

**Competing interests:** The authors have also filed patents relating to the FTC prodrugs and their combination with ETFD approaches to form SSPN therapy options. C.F. reports serving as a paid consultant for Cipla Pharmaceuticals, Janssen Pharmaceuticals, Merck Laboratories, Mylan Pharmaceuticals, and ViiV Healthcare, and received research grant support from Gilead Sciences paid to his University. The authors declare no competing interests.

