## [Peer Review File · Nature Communications]

Reviewers' comments:

Reviewer #1 (Remarks to the Author):

This manuscript reports the synthesis and in vitro characterization of semi-solid prodrug nanoparticles (SSPNs) for long-acting delivery of water-soluble antireoviral drugs (ARVs), in particular emtricitabine or FTC. The authors report the potential of emulsion-templated freeze-drying (ETFD) to develop libraries of FTC prodrugs suitable for long-acting depot administration, and characterize their potential pharmacokinetic (PK) profile using an in vitro in vivo (IVIVE) extrapolation model.

Adherence to daily ARV pill intake is a significant problem that reduces the effectiveness of current ARV therapies and prevention regimens. Long-acting injectables are, therefore, a highly desirable formulation that may potentially increase adherence and effectiveness of HIV drugs.

This manuscript reports valuable and relevant data.

Comments:

1. The simulated PK data for oral FTC seem to overestimate the clinical observed data. Would the PK parameters of the prodrugs estimated with this model be expected to overestimate the clinical data by up to 60-70%?
2. Although the need to space the injections is clear, 2g of formulation appear to be high. What would be the estimated volume that such an injection would require?
3. Recognizing the desire to reduce animal testing as well as the NC3Rs principles, the relevance of the findings reported in this manuscript would be enhanced if the simulated PK results and SSPNs properties were to be confirmed by a single animal study of the selected lead FTC C8 prodrug. The Authors state these studies are ongoing. Can at least part of those data be incorporated in this paper?
4. Have the process of synthesis and formulation changed in any way the antiviral activity of FTC? Have in vitro antiviral activity assays comparing the parent molecule with the prodrugs and the SSPNs been performed?
5. Based on the reported data, please comment about the feasibility of developing a 3 drug SSPN combination containing, for instance, tenofovir prodrug, FTC and dolutegravir, a first line regimen.
6. Please spell out IVIVE on line 32, as it does not appear to have been spelled out before.

Reviewer #2 (Remarks to the Author):

This manuscript details the development of a long acting drug formulation of HIV treatments. This is an exceptionally interesting pharmaceuticals study and accompanying modelling analysis to determine the likelihood of the formulation having pharmacological effectiveness in patients. The prodrug development approach taken here is highly innovative - most nanoformulation studies, whilst effective in animal models etc., are plagued by poor drug loading of the nanoparticle. Solid drug nanoparticles offer the opportunity to overcome this issue, but as shown here, the physicochemical properties of the drug(s) can limit the ability to make the formulations.

Given the novelty of the approach, the obvious clinical need, the fact that the formulation would

appear to have potential for further clinical development warrants publication in Nature Communications subject to some modifications and possible additions.

The main comments at this time that I have are essentially based around the clarity of the paper - although there is a substantial amount of data in the supplementary files, the paper itself is reasonably brief and more detailed explanations are needed in places.

Comments:

1. Line 65 - the dual nanoparticle approach is mentioned - but not explained - this is the first mention of a nanoparticle in the manuscript - this needs better clarification
2. The novelty and importance of the approach are somewhat subdued in the final paragraph - highlight the main features of the work that are to be covered better - ETFD is a means to an end here - not the main innovation
3. Figure 1 - the process of ETFD could be better revealed in the figure. Also, the figure implies release of the prodrug from the particle in the beaker at the moment - this could be expanded somewhat. The same is true for the legend
4. Lines 117/118 - I felt that the description of the library of resultant prodrugs could be slightly better described - it is confusing at the moment. It was not until careful examination of the supplemental figures that this became clear
5. Figure 2B - not clear what the blue band with capecitabine is - this needs more detail in the legend.
6. Figure 4 - this is a good way of summarising the factorial screen with excipients and surfactants. However, what does the diameter of the circles represent?

Reviewer #3 (Remarks to the Author):

1. This manuscript provides important information that may lead to FTC nanoformulation for HIV prevention and treatment.
2. Most of that data appear to reflect a single experiment, therefore reproducibility is not able to be determined.
3. No information is provided about the HPLC assay as far as accuracy and reproducibility.
4. Since FTC will need to be combined with another NRTI, the discussion should address how this will influence next steps in the FTC formulation strategy.

Reviewer 1 comments and responses

1) Please spell out IVIVE on line 32, as it does not appear to have been spelled out before.

Author response: We apologise for this mistake and have moved the later defining statement to this first use of the acronym IVIVE as correctly stated by the reviewer. The text now reads as:

“...aqueous nanodispersions that are predicted through in vitro to in vivo extrapolation (IVIVE) modelling to permit sustained release of prodrug...”

2) The simulated PK data for oral FTC seem to overestimate the clinical observed data. Would the PK parameters of the prodrugs estimated with this model be expected to overestimate the clinical data by up to 60-70%?

Author response: The ultimate quality of all modelling approaches is reliant upon the quality of the input data. The model used in the manuscript was generated using the best available literature parameters to describe the observed clinical data. In order to address the reviewer’s concerns regarding overestimation, the model has been refined by adjusting key parameters (such as plasma clearance) and was qualified against 3 data sets from clinical studies. These steps resulted in a closer matching of the observed clinical data and allowed a second prediction of the duration of therapeutic exposure. As the prodrugs are new chemical entities, accurate clinical data is not available for these molecules and it is currently not possible to accurately estimate the probable variation in PK of each prodrug; the simulated data from the second model suggests that while the plasma concentrations were lower using these parameters, the SSPN formulations still showed the potential to provide therapeutic intracellular concentrations of the active form

of FTC (intra cellular FTC-TP). The original modelling has been retained within the paper as this is based upon published descriptors; however, to address the reviewer's comments, this second ad hoc model has now also been included as supplementary information. Additionally, the following text was added to the manuscript:

"The simulations presented here were generated using input data determined experimentally, from literature values or predicted values and validated against a single clinical study. In order to investigate the predictive quality of the simulated data, key model parameters such as plasma clearance were modified and SSPNs 4 and 8 were again shown to have the potential to act as a viable long acting depot injection (Supplementary Methods and Supplementary Tables 8-11)"

3) Although the need to space the injections is clear, 2g of formulation appear to be high. What would be the estimated volume that such an injection would require?

Author response: The model assumes an injection volume of 6mL (the current maximum clinically used). In order to address the reviewers concerns a section discussing the usage of large volume IM injections has been added to the discussion, as follows:

"While the simulated PK of SSPNs are highly encouraging the simulated doses are high. Intramuscular injection volumes as high as 6mL have been administered previously, but smaller volumes would be preferable particularly when repeated administration is required. Recently, co-administration of hyaluronidase has been used as a strategy to reduce patient discomfort and enable higher volume subcutaneous injections. Hyaluronidase acts via hydrolysis of hyaluronan a key component of the extracellular matrix, resulting in increased membrane permeability. This approach has been used to increase the volume for both subcutaneous and intramuscular injections of anaesthetic agents, and subcutaneous administration of immunoglobulins."

4) Recognizing the desire to reduce animal testing as well as the NC3Rs principles, the relevance of the findings reported in this manuscript would be enhanced if the simulated PK results and SSPNs properties were to be confirmed by a single animal study of the selected lead FTC C8 prodrug. The Authors state these studies are ongoing. Can at least part of those data be incorporated in this paper?

Author response: The ongoing animal experimentation is being conducted with a complex experimental design aimed at assessing cross-species pharmacokinetics as well as pharmacodynamics in a humanized mouse model. The authors believe that this is beyond the scope of what could reasonably be included in the current manuscript.

5) Based on the reported data, please comment about the feasibility of developing a 3 drug SSPN combination containing, for instance, tenofovir prodrug, FTC and dolutegravir, a first line regimen.

Author response: A section discussing the potential for multiple drug LA IM formulations has been included.

“Current treatment strategies focus on treating HIV via a combination of antiretrovirals and the EFTD technology is readily compatible with generating combination nanoparticles comprised of more than one agent. This has previously been empirically demonstrated with lopinavir/ritonavir co-formulation and the presence of two agents within individual nanoparticles has been explicitly demonstrated using fluorescence resonance energy transfer (FRET) dyes. Future work will assess the potential to generate formulations containing multiple antiretroviral drugs.”

6) Have the process of synthesis and formulation changed in any way the antiviral activity of FTC? Have in vitro antiviral activity assays comparing the parent molecule with the prodrugs and the SSPNs been performed?

Author response: The primary goal of the prodrug work was to demonstrate that we can enhance lipophilicity sufficiently to promote SSPN synthesis, and to demonstrate that these prodrugs undergo enzyme-promoted activation to the parent drug FTC following prodrug release from the SSPN (similarly to capecitabine which is known to activate in the liver as opposed to target cells). The prodrugs are not themselves expected to display potent antiviral activity. This is based on knowledge of the mechanism of action of FTC and the known stability of carbamates. FTC is a cytidine analog that undergoes activation to the corresponding 5'-triphosphate and inhibits HIV reverse transcriptase, by acting as a dead-end CTP substrate analog. Thus, both the 5'-hydroxy and amine groups of FTC must be unmasked to exert antiviral activity. While it is conceivable that cleavage of the 5'-carbonate of FTC prodrugs can occur within target cells, we expect cleavage of the carbamate, also required for antiviral activity, to be prohibitively slow. Thus, we felt it was necessary to demonstrate that extracellular prodrug activation is achievable. From our work, it is clear that FTC carbamates are cleaved most efficiently in liver S9. We have modified the introduction to include prodrug design considerations and expected activity for clarity, as follows:

“Given the potential stability of carbamate functional groups in target cells, intact prodrugs are not expected to exhibit antiviral activity. However, the formation of stabilised prodrug nanoparticles, capable of releasing prodrug molecules that are activated under relevant biological conditions, could lead to candidates for HIV combination depot applications (Fig. 1).”

[Redacted]

Reviewer 2 comments and responses

1) Line 65 - the dual nanoparticle approach is mentioned - but not explained - this is the first mention of a nanoparticle in the manuscript - this needs better clarification

Author response: We apologise for the lack of clarity. The description of a “dual nanoparticle” approach referred directly to the combination of the two nanoparticle formulations (rilpivirine and cabotegravir) in the immediately preceding sentences. In order to avoid this confusion, the following words have been inserted into this sentence:

“This combination, representing a dual-nanoparticle LA option,…”

2) The novelty and importance of the approach are somewhat subdued in the final paragraph - highlight the main features of the work that are to be covered better - ETFD is a means to an end here - not the main innovation

Author response: We fully agree with the reviewer and have modified the text as follows:

“The SSPN formulations are predicted to deliver FTC at an appropriate plasma concentration to match dosing regimens estimated for the clinically-advanced cabotegravir and rilpivirine depot candidates. This offers the potential for water-soluble backbone NRTI delivery to supplement these industrial developments and increase the options for HIV patients globally; the combination of this SSPN LA option with the individual cabotegravir or rilpivirine LAI candidates, would immediately double the long-acting injectable options available clinically. The potential for long-acting NRTI delivery, through the alteration of the parent drugs towards hydrophobic and poorly water-soluble prodrugs, opens considerable scope for future therapy development for various diseases, especially as low melting-point materials have been highlighted as viable development candidates as SSPNs. Our ongoing studies aim to establish in vivo safety, pharmacokinetics and efficacy to support progression towards human evaluation and

create opportunities for additional patient benefits such as improved lifestyles, reduced pill burdens and long-term stability of adherence.”

3) Figure 1 - the process of ETFD could be better revealed in the figure. Also, the figure implies release of the prodrug from the particle in the beaker at the moment - this could be expanded somewhat. The same is true for the legend

Author response: Figure 1 has been modified to clarify the ETFD screening procedure and the Figure legend has been modified to read as follows:

“Figure 1. Schematic overview of long acting semi-solid prodrug nanoparticle (SSPN) strategy. Emulsion templated freeze drying (ETFD) screening is applied to a range of emtricitabine (FTC) prodrugs bearing bio-reversible carbonate and carbamate masking groups. SSPNs are released into aqueous media after rapid dispersion of ETFD monoliths into water, resulting in a dispersion of SSPNs slowly releasing prodrug under physiologically relevant conditions and subsequent bioactivation to release the parent FTC.”

4) Lines 117/118 - I felt that the description of the library of resultant prodrugs could be slightly better described - it is confusing at the moment. It was not until careful examination of the supplemental figures that this became clear

Author response: We are unsure where the lack of clarity has arisen in this part of the manuscript as the structures of the 16 prodrugs are in Figure 1 and they are labelled 1-16. We are hoping that the modifications to Figure 1 make this more clear.

5) Figure 4 - this is a good way of summarising the factorial screen with excipients and surfactants. However, what does the diameter of the circles represent?

Author response: We thank the reviewer for this positive comment and have added the following text to the figure caption to add additional clarity:

“Each colour represents a size range (as indicated by the legend) and circle diameter represents whether the z-average diameter value resides at the lower or higher end of the range.”

6) Figure 2B - not clear what the blue band with capecitabine is - this needs more detail in the legend.

Author response: In Figure 2, we have used the light blue band to illustrate the rate or half-life of carbamate cleavage of capecitabine in our HPLC assays. The center of the blue band indicates the mean initial rate or half-life, while the width of the band indicates the standard deviation in our measurements. To clarify this in the text, an additional statement has been added to the legend for Figure 2 as follows:

“The light blue band represents the mean \pm SD for carbamate cleavage from capecitabine.”

Reviewer 3 comments and responses

1) Most of that data appear to reflect a single experiment, therefore reproducibility is not able to be determined.

Author response: The reviewer is correct that the early stages of screening at 10 wt% prodrug were carried out rapidly to establish viability for higher drug loading materials. Multiple repeats were generated for SSPNs at ≥ 50 wt% drug loading. The following text has been added to the main body of the manuscript and Supplementary Table 3 to the Supporting Information:

“Accelerated screening with low quantities of drug compound (10 wt%) was used to provide clear guidance for options containing higher drug-loadings as these materials consume considerable amounts of prodrug compounds; options loaded at >50 wt% were subject to multiple repeats to establish robustness and reproducibility (Supplementary Table 3).”

2) Since FTC will need to be combined with another NRTI, the discussion should address how this will influence next steps in the FTC formulation strategy.

Author response: As the reviewer states FTC is not administered as monotherapy and we describe the proposed SSPN depot LAI as a supplement to the clinically-advanced rilpivirine and cabotegravir options, throughout the manuscript. To address the reviewers, and further clarify this point, a specific section discussing the potential for multiple drug LA IM formulations has been added to the discussion (see response to Reviewer 1 and modifications to the discussion to address comments by Reviewer 2).

3) No information is provided about the HPLC assay as far as accuracy and reproducibility.

Author response: We have included a new information in the Supporting Information under the “Kinetic Analysis” section, entitled “**HPLC method for analysis of prodrug cleavage**” that describes the HPLC method in detail and includes representative HPLC data (Supplementary Figures 17, 18, 20 & 22-23; Supplementary Table 1). The HPLC method developed for the reported prodrugs provides adequate separation between prodrugs and observed products. Prodrug concentrations evaluated using this method are confirmed by comparison to a standard curve. Retention times and UV profiles of prodrugs and products are confirmed by comparison to authentic standards. HPLC analyses to determine initial rates were performed in triplicate to demonstrate the reproducibility of the assay. In addition, the following test has been added to address the question of accuracy and reproducibility

“Compounds 8 and 16 were utilized to determine inter- and intra-day accuracy and precision: at the high and middle points of the standard curve these were below 15%; at the low point of the standard curve for both compounds, inter- and intra-day accuracy and precision were within 20% in accordance with convention”

Editorial Note: Reviewer #1 was unable to assess this round of review and so the other reviewers, who have overlapping expertise with reviewer #1, were asked to comment on the responses given to reviewer #1.

REVIEWERS' COMMENTS:

Reviewer #2 (Remarks to the Author):

The authors have responded to my queries very well and the manuscript is much more clarified. I believe that the responses to the other comments from the other reviewers is also very helpful.

Reviewer #3 (Remarks to the Author):

The responses to the reviewers comments and the revised manuscript have addressed prior concerns.